# Two Sides of the Same Virtual Coin: Investigating Psychosocial Effects of Video Game Play, including Stress Relief Motivations as a Gateway to Problematic Video Game Usage

**DOI:** 10.3390/healthcare12070772

**Published:** 2024-04-02

**Authors:** George Farmer, Joanne Lloyd

**Affiliations:** 1Westminster Centre for Psychological Sciences, University of Westminster, London W1W 6UW, UK; 2Cyberpsychology Research—University of Wolverhampton, School of Psychology, Faculty of Education, Health and Wellbeing, University of Wolverhampton, Wolverhampton WV1 1LY, UK; joanne.lloyd@wlv.ac.uk

**Keywords:** video games, stress, motivation, problematic gaming, stress relief, behavioural addiction

## Abstract

Video gamers can play to negate the psychological impact of stress, which may become problematic when users over-rely on the stress relief potential of gaming. This study used a repeated measures experimental design to investigate the relationships between stress, video gaming, and problematic video gaming behaviours in a convenience sample of 40 students at a UK university. The results indicated that positive affect increased and negative affect decreased, whilst a biological stress measure (instantaneous pulse rate) also decreased after a short video gaming session (*t*(36) = 4.82, *p* < 0.001, *d* = 0.79). The results also suggested that video gaming can act as a short-term buffer against the physiological impact of stress. Further research should focus on testing individuals who have been tested for gaming disorder, as opposed to the general population. Research could also utilise variations of the methodological framework used in this study to examine the intensity of a stress relief effect under different social situations. The study’s findings in relation to published works are also discussed.

## 1. Introduction

Video gaming is a ubiquitous technology that has become a part of everyday life. In the UK alone, consumers spent roughly GBP 4.75 billion on video games in 2022, with the video game consumer market being worth GBP 7.05 billion [1]. This has followed a rapid growth of interest in gaming following the COVID-19 pandemic [2], during which video games played a crucial role in mitigating experiences of stress and other prominent psychological health concerns during a time of intense social isolation [3,4,5,6].

However, the growth of video game consumption has piqued the interest of academia: the debate surrounding video games and their social consequences gravitates around two central ideals. On one side, some articles suggest that video games can be utilised for positive psychosocial outcomes such as increasing social connectivity, improving psychological well-being, and encouraging cooperation between players [7,8,9]. On the other side, some articles suggest that video games have the potential to increase aggressive cognitions [10,11] or may be used in problematic ways that can evolve into a behavioural addiction if used in a way that severely disrupts facets of everyday life [12,13,14].

Video gaming also has a versatile range of applications in healthcare settings: e.g., using video games as a “distractor activity” allowed children undergoing chemotherapy to regain a sense of resilience to the harsh side-effects of the therapy [15]. Engagement in video gaming also helped to increase coping skills, internal locus of control, and self-management of symptoms by playing minigames as a superhero, including metaphors for beating cancer and ultimately empowering the patient to fight through depressive symptoms [15]. It has also been used in a traditional therapeutic context, in order to bridge the gap between therapist and client, a solution that is particularly effective with younger clients [16,17,18].

Outside of clinical contexts, having an online presence within a video game can help to promote the cultivation of strong relationships with other players through shared experiences [19]. Indeed, many choose to keep playing these types of games on the hope that they themselves will partake in such rich social interactions with other players [20]. In-game social interactions can provide an unorthodox therapeutic outlet otherwise unavailable to players, and many players find that in-game experiences are not available to them offline [21]. It is the availability of diverse emotional experiences that have the potential to elicit complex psychosocial outcomes, such as improved mood and psychological regulation of stress [22,23,24].

However, extreme exposure to stress offline may drive some to coping mechanisms through widely accessible means, such as technology [25,26,27]. It has been widely discussed that escapism-driven uses of video games especially can lead to maladaptive use behaviours if the player becomes dependent on the psychological benefits of these activities to alleviate stress [28,29]. Furthermore, Dutcher and Cresswell [30] highlight the role of dopaminergic reward pathways in the regulation of stress. Given that these pathways also feature in the development of behavioural addiction [31], it is important to learn more about the relative benefits and risk of activities such as gaming, which have the potential to be both stress-relieving and addictive.

Reinecke [27] demonstrates that using interactive media such as games has a “significant recovery potential and are frequently used after stress and strain for recovery reasons” (p. 26). Furthermore, this research finds that work-related stressors are a reliable predictor for the usage of video games, and that individuals can adapt their use of video and computer games to their individual circumstances. The research covered so far demonstrates that video game players walk a fine line between enjoyable and problematic uses of games, depending on their primary motivations for use.

Some video game players exhibit symptoms of video game addiction, which has been clinically recognised in previous literature and the criteria for diagnosis published within the International Classification of Diseases, eleventh edition [12,32,33]. The clinical criteria for gaming disorder include losing control over the amount of time spent playing games, conflicts with friends and family members over their habits, and the increased prioritisation of gaming, as opposed to work or academic performance [12,34]. Previous research has attributed this rise in addiction to gaming, in part, on the failure to either recognise emotions (alexithymia) or manage emotions, using video gaming to experiment with and correct emotional availability [35]. However, it may be that a small minority of video game players are inefficiently regulating mood or over-relying on the game to regulate their mood for them, which has been observed in other modes of technology, such as mobile phone apps [36].

Within the video game psychology literature, there are several research papers that investigate the role of video game usage as a way of creating and promoting emotional regulation strategies [37]. Emotional regulation is the process of people achieving targeted changes or fluctuations in their mood by engaging with specific behaviours, leading to changes in positive and negative affect. It is posited by Villani and colleagues that people may use video games as a way of “enhancing their emotional lives and protecting themselves from psychopathologies” (p. 2). It could be plausible that video gaming is being used as a psychological “mood enhancer” and may explain why it has the potential to become a cathartic relief to the stresses of everyday life.

Previous research suggests that individuals who have problems with emotional regulation are more likely to engage in addictive behaviours to escape from, or minimize, negative moods [38]. It is possible that players who adopt maladaptive coping strategies (e.g., increased play time, neglect of social responsibility, etc.) become dependent on positive video game effects such as psychological need satisfaction [39] or mood regulation [40] to escape from and regulate the negative psychological impact of offline stress.

However, it has become a source of debate over whether behaviours such as gaming and problematic Internet usage should be classified as an addiction in the same way as substance disorders [41,42]. It has been suggested that these share similar characteristics in terms of tolerance, mood modification, relapse rates, and withdrawal symptoms [43]. However, alternative theoretical frameworks could be better suited to explain Internet gaming disorder.

Kardefelt-Winther responded to the discourse on Internet addiction by suggesting that Internet usage could be explained as a compensatory mechanism [44]. In theory, users experience a lack of social resources (e.g., social capital or cognitive arousal) offline, and therefore increasingly depend on the Internet and the online stratosphere to provide experiences that provide the resources which they lack. This is opposed to an addiction framework, as the compensatory usage model conceptually defines problematic usage as a way of fulfilling a need in a highly engaging way or can develop as a result of a maladaptive response to stress [29]. This particular model highlights the importance of stress as a unique risk factor, as opposed to the myriad of other factors commonly associated with Internet addiction, such as personality traits [45] and psychological well-being [46].

This study investigates the impact of a short-term video gaming session upon both biological and self-reported stress levels. Stress has been used in previous work as an ecologically valid way of measuring affective change to repair mood and prevent negative physiological consequences [24]. If the motivations of users are to decrease negative affect and increase positive mood, we hypothesise that it would be reflected in decreased stress scores. Whilst Kardefelt-Winther established this effect in the domain of Internet usage, it will be the purpose of this study to test whether these effects are applicable to video game players. If these effects can be demonstrably exhibited in video game players, it would have implications on how video gaming can be used in both “healthy” and “unhealthy” ways.

This research is based on the theory that emotional regulation strategies through video game play act as positive psychosocial compensation, which for moderately engaged players can be a healthy coping strategy [35]. However, it would also be of interest to investigate potential correlations between more problematic video game use and mood shift after video game exposure, as a way of clarifying possible motivations for video game playing in those who score highly on Internet gaming disorder (IGD) criteria [47]. To investigate these assertions, an experimental task will be employed to compare stress before and after exposure to a commercial video game, with the expected effects of positive affective changes after video game exposure. It is unknown whether a potential stress-reduction effect after playing a video game would contribute towards the development of problematic gaming behaviours; however, we expect that this may be the case, based on the principles of compensatory internet use described by Kardefelt-Winther [44].

An Internet gaming disorder (IGD) scale will be used to assess self-reported levels of gaming disorder in participants. In accordance with Kardefelt-Winther’s research on Internet addiction [44], which demonstrated a mediated link between problematic internet use and affect, it is expected that those scoring highly on problematic gaming behaviours will also exhibit a higher affect shift. This will therefore be assessing whether those who register positive for several IGD criteria experience an elevation in mood and decrease in stress, which may be able to explain why players show problematic patterns of behaviour. Regardless of the motivations for play, we expect to find that video game play has a positive physiological impact on players. This is supported by previous literature that suggests single video gaming sessions can have positive effects on players [7,9].

To summarise, this research will be investigating the correlations between stress (at both a biological and a self-reported level), video gaming, and (self-reported) problematic video gaming behaviours, grounded in Kardefelt-Winther’s compensatory Internet use theory [44]. This introduction has identified several key areas within the extant literature of video gaming psychology, which we feel provides a much-needed contextual snapshot of how the psychological benefits of video gaming could reinforce an unhealthy dependence on positive state shifts. Given the wealth of research currently dedicated to addiction models of video game play, the following research study focuses more specifically on an experimental test of stress reduction and mood management after playing a video game. This is important to explore, as it could potentially be one mechanism contributing to the development of problematic behavioural patterns, but the exploration of whether this effect directly contributes to problematic/disordered gaming per se is beyond the scope of this study.

Considering the evaluation of relevant literature and the aims of the study, we predict that the following hypotheses will be true:H1—Exposure to video game play will decrease biological measures of stress.H2—Exposure to video game play will improve positive affect scores.H3—Exposure to video game play will reduce negative affect scores.H4—Higher scores on a gaming disorder scale will be associated with a greater increase in positive affect.H5—Differences in biological stress scores after video game play compared to before video game play will be associated with higher gaming disorder scores.

## 2. Methods

### 2.1. Participants

The data in this study were collected from a student sample at the University of Wolverhampton, and the participants were recruited through the advertisement of the study on a “Psychology Participant Pool” in exchange for course credits. The study had no inclusion criteria that focused on prior experience of video gaming. Exclusion criteria included anyone under the age of 18, those with a background of photosensitive epilepsy or migraines, and those without normal (or corrected to normal) levels of vision.

### 2.2. Measures and Technology

#### 2.2.1. Positive and Negative Affect Scale (PANAS)

State affect was measured using the positive and negative affect scale (PANAS; [48]) which was administered pre-game and post-game in order to measure the direct effect of the game on self-reported emotion. A widely used mood questionnaire, this scale is 20 items long, measuring both positive affect (10 items) and negative affect (10 items), tasking the participant to rate how they feel at the time of questioning using chosen adjectives (such as “excited” or “upset”) on a Likert scale of one–five (one = “very slightly or not at all [adjective]”; five = “extremely [adjective]”). The mood scores are then tallied at the bottom of the questionnaire, resulting in a personalised positive and negative mood score out of a possible 50 points for each variable. Internal reliability scores for this scale have been reported between *α* = 0.83 and *α* = 0.90 for positive and negative affect [49]. Using this scale provides a reliable measure of mood and affect “in the moment”, which was a logical choice for a repeated measures experimental study and allows for a direct comparison across time points to observe the affective impact of the video game.

#### 2.2.2. Perceived Stress Scale (PSS)

Stress was measured using the perceived stress scale [50], a 10-item questionnaire asking the participants to rate how often they felt under stress in the last month, asking about certain scenarios illustrated in the questions (“how often have you felt that you were unable to control the important things in your life?”), using a Likert scale from zero (“never”) to four (“very often”). Questions 4, 5, 7, and 8 are reverse-scored to prevent order bias. This results in a score ranging from 0 to 40; the higher the score, the higher the self-reported stress level. Internal reliability scores have been reported within acceptable parameters for academic research in previous research articles ([51]; *α* = 0.83). Such measures of perceived stress are an easy-to-use and reliable method of discerning stress levels in participants. However, relying on self-report measures alone may have increased the possibility of measurement bias (c.f., [52]).

#### 2.2.3. Internet Gaming Disorder Scale 9—Short Form (IGDS9-SF)

Problematic gaming was measured by using a modified version of the Internet gaming disorder scale (IGDS9-SF; [53]). The version used in this experiment included the question “Have you experienced serious problems at work or school because of gaming?”. The item was included to reflect the diagnostic criteria that players continue to play despite adverse social consequences [47].

This 10-item questionnaire asks the participant to respond either “yes” or “no” to questions such as “Have you hidden the time you spend gaming from others?”, with a maximum score of 10. Any participants scoring 5/10 or above would be considered a problematic gamer, which has been adjusted from the original score used in previous work [53]. Internal reliability scores for the IGDS9-SF were reported to be the highest when compared to five similar gaming disorder scales ([54]; *α* = 0.89). Using this scale allowed the researchers to determine whether mood changes would correlate with problematic gaming behaviours.

#### 2.2.4. Instantaneous Pulse Rate (Photoplethysmography; PPG)

A biological level of stress was measured using instantaneous pulse rate (IPR; [55]), an alternative methodology to heart rate variability (HRV; [56]). If the pulse rate is higher than baseline levels, blood flow increases towards the wrist, whereas lower pulse rate indicates decreased blood flow to the wrist, which is a suitable comparison to a rise or fall in heart rate. If the pulse rate is higher than baseline, much like heart rate, it is indicative of higher stress levels. Pulse rate was taken using photoplethysmography (PPG). The amount of light absorbed and reflected by tissue in the wrist is regulated via the flow of blood in corresponding vessels, which represents biological indicators for behaviour marked by heart rate [57].

By scattering infrared light into the finger, the device can measure how much light is absorbed by the red blood cells, and therefore the level of red blood cells in the finger. The photo-cellular “cap” on the user’s fingertip converts light to electrical energy, which is measured as a mean value of heart rhythm between intervals. It measures pulse volume or phasic changes, which are related to beat variations in the force of blood flow. These beat-to-beat changes in peripheral blood flow reflect the heart’s inter-beat intervals, similar to electrocardiogram (ECG) methods. Nevertheless, this should not be confused with pulse rate variability, which measures changes around the mean and is not an estimate of IPR [55]. Using a biological measure of stress adds a greater level of scientific rigor to the experiment, rather than relying on self-report methods alone.

#### 2.2.5. Video Game

For the purposes of the study, Mario Kart 8 Deluxe (Nintendo) was used, with a Nintendo Switch console (Nintendo, Wolverhampton, UK). Mario Kart 8 Deluxe is a racing game that requires players to choose a character from previous Mario games and compete in go-kart races. This title was chosen for several key reasons—that a large number of people are already familiar with the Mario franchise as a commercially sold game [58]; that Mario Kart is particularly easy to play, even for those with little gaming experience; the game features balanced design choices [59]; and the console has a high overall satisfaction rating among users, regardless of prior level of gaming experience [60].

Whilst the game has multiplayer functionality, the gaming session used the “Grand Prix” mode on a single-player basis. This mode encompasses four races played sequentially on different levels for each race, lasting anytime between 20–30 min, depending on participant skill level.

### 2.3. Procedure

Prior to the beginning of the experiment, informed consent was gained, and demographic information such as age, gender identity, and previous gaming experience was then collected. The experimental task began with participants answering questionnaires measuring self-reported positive and negative affect levels, self-reported stress, and self-reported problematic gaming.

Once completed, the participant was informed that their heart rate would be taken and instructed to put the finger cap on. The participant was told that the researcher would note the score displayed on the device once every 30 s for a five-minute period. In three cases, the participant was unable to use the pulse oximeter and excused from the exercise, instead continuing to the video game phase of the experiment.

The researcher instructed participants on how to use the controls for the game and, for those with no experience, how to play. The participant was informed that after the fourth race was over, they would stop playing the game. Just before the fourth race started, the researcher informed the participant that it was the last race. This allowed the participants to come to a natural conclusion of play time, rather than abruptly stopping video game play, which could affect mood.

Immediately after video game play, the participant completed the pulse rate measurement for a second time or went straight to the final questionnaire if measurement was not possible. The PANAS questionnaire was administered a second time to measure any fluctuations after video game play. The participant was debriefed and any questions about the experiment or study were answered by the researcher.

#### Ethical Approval

Ethical approval was provided by the University of Wolverhampton Faculty of Education Health and Wellbeing ethics committee.

### 2.4. Research Design

This study used an experimental repeated-measures design. Self-reported levels of gaming disorder were the independent variable, whereas positive affect, negative affect, and a biological measure of stress (PPG) were the dependent variables.

## 3. Results

An a priori power analysis was conducted using G*Power3 [61] to test the difference between two dependent means (matched pairs) using a one-tailed test, a medium effect size (*d* = 0.50), and an alpha of 0.05. The results concluded that a total sample of 36 participants (*n* = 36) was required to achieve a power of 0.90. An a priori analysis was also conducted to determine the necessary power to detect medium effects (ρ = 0.30), given an alpha of 0.05 and power of 0.90, for correlation-based analyses. The results showed that a sample size of 88 (*n* = 88) would be necessary to power correlational testing that could detect medium effects. All inferential analyses were conducted using JASP 0.18.1.0 [62], an open-source statistical software package with a graphical user interface that features the ability to run commonly used statistical analyses including *t*-tests, ANOVAs, and regressions using both classical and Bayesian methods [63].

### 3.1. Demographic Results

The sample survey respondents (N = 40) had a range of ages (M = 24.73, SD = 7.61) between 18 and 61, and an almost even split between male and female participants (male = 18; female = 22). On an anecdotal level, participants reported a moderate level of gaming experience, varying from no experience to experienced players of video games, prior to the experiment.

### 3.2. Normality Testing

Shapiro–Wilk tests were conducted on all analyses, to test for violations of normality assumptions within the data set. Normality assumptions were not violated for the tests described in Section 3.4.1, Section 3.4.2, or Section 3.4.5 (*W* = 0.98, *p* = 0.82; *W* = 0.97, *p* = 0.54; *W* = 0.94, *p* = 0.054), suggesting that parametric analysis would be suitable for testing H1, H2, and H5. However, for the tests described in Section 3.4.3 and Section 3.4.4, normality results were statistically significant (*W* = 0.86, *p* < 0.001; *W* = 0.94, *p* < 0.05), suggesting that non-parametric analysis would be necessary to test H3 and H4.

### 3.3. Descriptive Statistics

Participant responses to the demographic items have been collated in the table below, including Means and Standard Deviations (see Table 1). Prior to inferential-level analysis, the variables of interest were subject to a preliminary correlation coefficient analysis to determine whether any statistically significant relationships were present within the data (see Table 2).

### 3.4. Hypothesis Testing

#### 3.4.1. Instantaneous Pulse Rate

To test H1, a matched pairs *t*-test concluded that participants had a decrease of 4.7 ms (SE: 0.98) in pulse rate on average following a short video gaming session, and that this decrease was statistically significant (*t*(36) = 4.82, *p* < 0.001, *d* = 0.79), as illustrated by Figure 1.

#### 3.4.2. Positive Affect

To test H2, a matched pairs *t*-test confirmed that there was an increase in positive affect scores by 3.32 points (SE: 0.91) on average per participant. This was identified as a statistically significant increase in scores after a short video game session (*t*(39) = 3.62, *p* < 0.001, *d* = 0.57), as illustrated in Figure 2.

#### 3.4.3. Negative Affect

As normality assumptions were violated (see Section 3.3), to test H3, a Wilcoxon signed ranks test was conducted. The results showed that negative affect scores before the video game session (*M* = 13) decreased by a small, but statistically significant, amount after exposure to the video game (*M* = 11; *W* = 502.5, *p* < 0.001, *r_B_* = 0.68). This is illustrated in Figure 3.

#### 3.4.4. Positive Affect and Problematic Gaming

A Spearman’s correlation indicated that there was not a statistically significant correlation between the difference in positive affect scores and problematic gaming scores (*ρ* = −0.07, *p* = 0.62). These results suggest that H4 was not supported at a statistically meaningful level.

#### 3.4.5. Instantaneous Pulse Rate and Problematic Gaming

A Pearson’s correlation coefficient was conducted to assess the association between the difference in pulse rate scores after exposure to video game play and self-reported IGD scores. The analysis produced a non-significant correlation (*r* = −0.15, *p* = 0.38), suggesting that H5 was not supported at a statistically meaningful level.

## 4. Discussion

The aim of this study was to test hypotheses, based on previous findings, that video game play would decrease biological indicators of a stress response (H1), encourage positive mood states (H2), and regulate negative mood states (H3). It was also expected that the degree of increase in positive mood states after gaming would be associated with higher scores on a gaming disorder questionnaire (H4), and that there would be an association between a biomarker of stress and gaming disorder scores (H5), providing support to compensatory use accounts of video game play and illustrating why video gamers may engage in problematic use behaviours. This research used an experimental research design, featuring widely accessible technology to measure biological markers of stress and a popular video game that would be accommodating of any game player, regardless of previous gaming experience. This study also provided a unique methodological perspective of measuring both biological and self-report measures of stress to partially mitigate some criticisms of biases involved in these techniques.

The results of this study suggest that video gaming has a measurable effect on biological stress, when isolated to a short gaming session within a laboratory environment. Exposure to the video game decreased stress levels compared to pre-experimental levels, which supports H1. This also supports a growing evidence base from previous literature that suggests video game play has been observed to measurably decrease experiences of stress [24,27,64,65]. Indeed, in a casual form, video gaming has been previously compared to guided relaxation or meditation [66].

The results suggest that the video game had a measurable effect on participant mood states. Self-reported positive mood scores increased by roughly three points per participant on average after the video gaming session, whereas self-reported negative mood scores decreased by roughly two points per participant on average after the video gaming session. The results therefore support H2 and H3.

The results did not support the prediction that problematic gaming scores would be associated with a more pronounced changed in affect as a result of playing the game, which was unexpected and does not support H4. The results also did not support the prediction that there would be an association between the difference in IPR scores measured after exposure to video game play and gaming disorder scores, which refuted H5.

One possible explanation for this is that the data set used for analysis did not have sufficient power to identify small effect sizes. Whilst an a priori power analysis indicated that the study was sufficiently powered to detect medium effect sizes, it is possible that the predicted relationship is only evident when measuring for smaller effect sizes, as discussed in previous literature on video game research [9]. Any attempts at replication would require a sample size of at least 97 based on the parameters described in Section 3 (for *t*-tests; *d* = 0.3 to account for smaller effect sizes), and 88 for correlational analysis, which was beyond the logistical scope of the study at the time. Therefore, we encourage future efforts to replicate the study with a larger sample.

Whilst it may be possible that video gaming’s ability to moderate affect plays a role in problematic gaming behaviours, the present study did not detect a disproportionate effect of gaming on the affect of those with higher IGDS9-SF scores. It must also be noted that lower scores on the IGDS9-SF are likely to have captured individuals that both do not participate in video gaming at all, as well as individuals that do participate in video gaming but play in an adaptive way to avoid the negative effects illustrated in the scale.

Two confounding variables that could explain these results may have been the difficulty level of the game or the experience levels of the participants. The “Grand Prix” game mode was set to “Easy” as a default option, to cater for a variety of experience levels in the target population. However, one might expect that individuals that exhibit higher levels of problematic gaming would be more experienced or skilled at the game used in the present study; for some participants, it was observed that this setting represented little-to-no challenge. Whilst these individuals may experience a small increase in mood, it is plausible that any mood increase would be attenuated by the easiness of the game by frustrating the players’ ability to experience achievement or competence from the gaming session.

This may also provide an explanation for the lack of association between IPR difference scores and gaming disorder scores (H5); individuals who exhibit higher levels of problematic gaming may be “desensitized” to a short video gaming session that provided little to no challenge—it may be the case that excitation of the nervous system (such as increased heart rate) simply would not occur unless certain goals are being met within the gaming session. This has been referred to as ‘gaming tolerance’ in previous literature [67]; however, there is a need to emphasise that this is more than players needing more time to engage with a video game. Indeed, it appears that there are several complex emotional and motivation-based factors to consider for those with gaming disorder, such as craving, fear of missing out, and the intense need to fulfil psychological needs [67,68,69].

For inexperienced players, an easy game allowed for a greater possibility of success, which may have improving feelings of competency, which has been associated with levels of psychological well-being [70]. However, the opposite was also true for more experienced players, which may have unintentionally created diametrically opposed ceiling and floor effects on the PANAS. It may also be the case that participants with less gaming experience would naturally experience more negative affect as levels of competency decrease by having to ask for instructions at an increased rate or experiencing feelings of helplessness. Nevertheless, if the study was replicated with the inclusion of controls for video game difficulty and prior gaming experience, it may be easier to observe the effects predicted in H4. Future attempts at replicating this study or improving upon it should consider the role of participants’ experience of video gaming prior to experimentation, as well as the challenge (i.e., difficulty) of the game itself.

This may be explained through the lens of self-determination theory [71], which suggests that individuals engage in behaviours that encourage the fulfilment of three basic psychological needs—autonomy, competence, and relatedness. As the video game fulfilled all or some of the three psychological needs, participants may have experienced a greater positive affect. As these needs may have been frustrated by a lack of experience with the game, participants may have experienced a greater negative affect shift. However, the degree to which this was experienced and whether it can truly be asserted that experience would be a significantly moderating factor as to shift self-reported affect scores remains to be seen. Further research should endeavour to explore the nature of the relationships between the variables observed and theorised in this study.

The changes in mood in this experiment imply that video games can have a short-term stress relief effect; this is similar to the theoretical principles of compensatory internet use, examined by Kardefelt-Winther [44]. It could be suggested that problematic gaming behaviours are driven by the need to attenuate the effects of stress by interacting with video games, ameliorating this stress for a short while. It could be in this small time zone in which the overall experience of video gaming remains positive; it could be argued that the continued use and therefore over-reliance of gaming effects is what contributes to negative experiences and potential addiction [72].

The results of this study implies that anyone wishing to experience a stress relief impact from video gaming should already be at a moderately stress-free cognitive stage [73]. This also provides some explanation as to why problematic gaming has such a strong association with stress [74,75]; the compensatory Internet use theory [44] posits that external social stressors contribute to an increased investment in resources (time, etc.) by gaming more to escape the negative consequences of this stress. However, it has been theorised that escapism is only effective as a short-term method of stress relief (by dissipating some stress), with the potential to become both a new source of stress, and individuals developing an addictive behavioural relationship with gaming as a result [74].

### 4.1. Limitations and Considerations

A limitation of this study is that the experiment was conducted without a control group, so it is not possible to be certain whether the mood state effects observed were solely due to the influence of the video game. Indeed, a control group would have established whether changes in mood or biological stress were influenced by any natural relaxation effects as the participants became more comfortable with the laboratory environment and could have used an unrelated activity as a comparable measure. However, the experimental protocol did partially account for relaxation effects by allowing participants to “settle in” to the laboratory space, allowing for a short amount of time in which the participant’s heart rate would normalise before the first set of PPG readings. Enforcing a brief rest period before readings were taken allowed for a more natural baseline heart rate measurement in participants and would potentially mitigate any environmental stressors.

Previous research with a similar research methodology found that changes in affect measured by the PANAS in the gaming condition were larger, whereas the control condition reported only mild changes in affect [66]. It appears that whilst the results in the present study may have captured a relaxation effect occurring, the positive affect shift was more pronounced than in a comparative study. Regarding stress, previous research has established that video games, regardless of content, are able to reduce stress [76,77], which the results of the present study further supports. However, previous research suggests that biomarkers of stress increased slightly from pre-gameplay levels to post-gameplay levels (HRV; [56]), which was not observed in the present study. This suggests that the results of the present study go beyond an expected general relaxation effect and that a short session of video gaming may provide tangible stress relief benefits for the player.

One limitation of this study is the use of a student sample. Whilst this was convenient for the fulfilment of the study aims, the use of a more heterogeneous sample may be more impactful to the wider discussion of video games research. The results of this study provide further insights as to the video gaming behaviours of university students but are not necessarily generalisable to a wider population. While the study did not explore gender as a variable of interest, some studies have found interesting gender effects in how people engage with and benefit from video games [78], and it would be valuable for future research to further investigate whether there are gender differences in stress responses to video gaming.

Furthermore, the results have somewhat limited ecological validity. Although we used a widely available commercial game and console to replicate a real-world gaming session as closely as possible, having the video game provided to participants, rather than the participants choosing a game out of preference, may mean that the findings of the study do not fully reflect the actions, experiences, or any potential stress-reduction effects that might occur in more naturalistic conditions. However, using a video game that was actively chosen by the participant would have represented a considerable variation to the design of this study and whilst this could have increased the ecological validity of the experiment, the researchers chose to impose an increased level of experimental control instead. The presence of a stress-reduction effect in laboratory-based conditions suggests the robustness of such effects, and we welcome the publication of further research using more naturalistic settings and other methodologies.

Another limitation of this study was the length of time used to play the video game chosen. Whilst the “Grand Prix” mode was appropriate to experience a wide range of aesthetic or enjoyable experiences within the video game, it is unclear whether a 20–30 min session of video gaming is sensitive enough from a methodological perspective to identify any significant stress relief or mood regulation changes compared to before the session. However, a systematic review of video gaming for the relief of stress and anxiety noted that even one-to-five-minute sessions of gameplay were effective at reducing stress [65], which was supported in this study. Therefore, duration may not be a major limitation to observing stress–health relationships.

Additionally, while the experiment did not measure players’ experiences of the flow state (c.f., [79]), which may explain facets of player enjoyment and mood while gaming, it would be of interest to include this variable in any future replication efforts. Future experimental designs in this field may want to explore the role of gestures, voice, and other physical parameters that would contribute to the expression of emotion that would be controlled by the act of video gaming, as highlighted in previous research [80].

Finally, this study used PPG as a measure of instantaneous pulse rate, which is an alternative to materials used to measure heart rate variability. This was chosen based on accessibility and convenience as the author did not have access to electrocardiogram (ECG) equipment that produces the signal necessary to measure HRV. It is acknowledged that whilst the signal used during IPR measurement “exists between ECG and PPG signals” [81], it is not the most ideal proxy for measuring heart rate. Using a measure such as instantaneous pulse rate variability [82,83] may have been a better alternative considering the resources available. For future research or replication efforts, where resources allow, using ECG or HRV measures would be considered the gold standard for biological measures of stress.

### 4.2. Conclusions

In conclusion, video games have the potential to reduce biological measures of stress, whilst generally improving mood states (lowered negative state, higher positive state), after a short session in experimental laboratory conditions. Whilst this study cannot make any conclusive claims as to how this relates to the initiation or maintenance of problematic video game behaviours, the results support previous literature that asserts video gaming as comparable to alternative methods of stress relief [65]. The observed changes in mood state and stress have implications for wider health effects, mainly that video games have the potential to positively influence physical and psychological well-being in short durations. The findings of this study also provide useful information for those working with people experiencing symptoms of gaming addiction, or for parents or educators attempting to regulate young people’s engagement with video games, as it provides insights into one of the reasons why people may seek out gaming at times of stress. It follows that those attempting to cut down their gaming, particularly when struggling with stress or low mood, would benefit from support or encouragement to identify alternative methods to manage their stress. However, the findings are limited by some methodological design flaws and would benefit from replication with the addition of a control group, and the addition of variables such as prior experience of video gaming, flow, and physical expressions of emotion.

## Figures and Tables

**Figure 1 healthcare-12-00772-f001:**
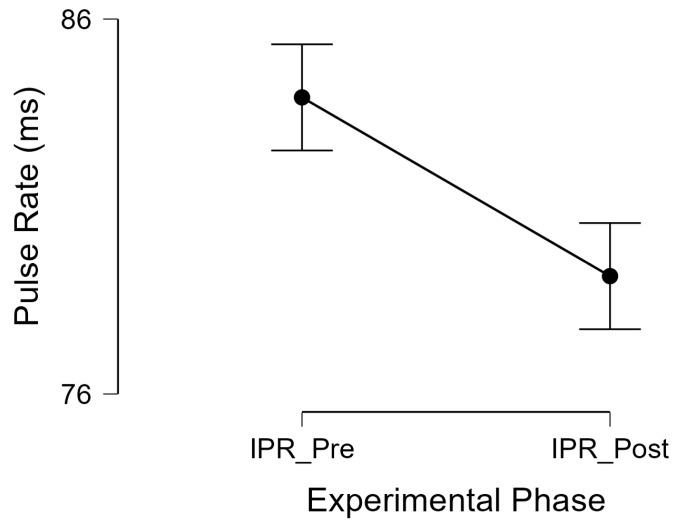
Pulse rate decreases after a short video game session. Note: IPR_Pre = instantaneous pulse rate pre-videogame; IPR_Post = instantaneous pulse rate post-videogame.

**Figure 2 healthcare-12-00772-f002:**
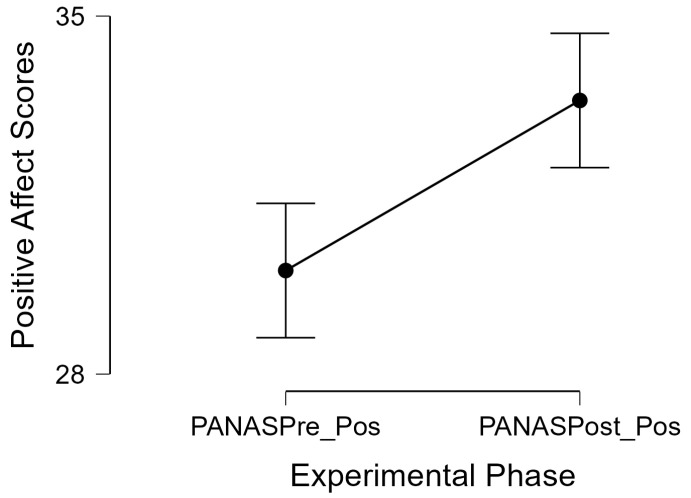
Positive affect increases after a short video gaming session. Note: PANASPre_Pos = PANAS positive affect score pre-videogame; PANASPost_Pos = PANAS positive affect score post-videogame.

**Figure 3 healthcare-12-00772-f003:**
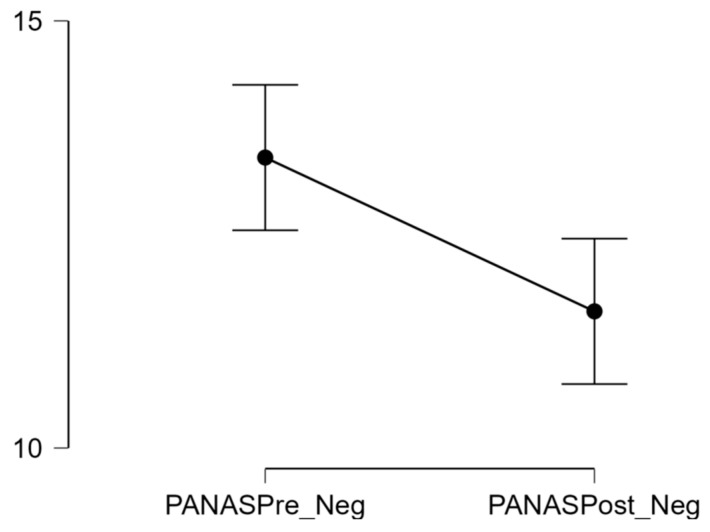
Negative affect decreases after a short video game session. Note: PANASPre_Neg = PANAS negative affect score pre-videogame; PANASPost_Neg = PANAS negative affect score post-videogame.

**Table 1 healthcare-12-00772-t001:** Descriptive Statistics.

	N	Mean	Std. Deviation	Range	Minimum	Maximum
Age	40	24.73	7.61	43.00	18.00	61.00
PANASPre_Pos	40	30.02	7.87	38.00	10.00	48.00
PANASPre_Neg	40	13.40	3.56	18.00	10.00	28.00
IGD_10	40	2.95	1.83	9.00	0.00	9.00
PSS	40	19.38	5.13	16.00	12.00	28.00
IPR_Pre	37	83.91	10.83	41.83	63.00	104.83
IPR_Post	37	79.14	9.70	38.50	60.83	99.33
PANASPost_Pos	40	33.35	7.88	33.00	14.00	47.00
PANASPost_Neg	40	11.60	3.34	17.00	10.00	27.00
IPR_Diff	37	−4.77	6.01	25.67	−18.34	7.33
PANASPos_Diff	40	3.33	5.81	27.00	−11.00	16.00
PANASNeg_Diff	40	−1.80	3.76	26.00	−15.00	11.00

Note: PANASPre_Pos = positive affect before game play; PANASPre_Neg = negative affect before game play; IGD_10 = Internet gaming disorder; PSS = perceived stress; IPR_Pre = instantaneous pulse rate pre-videogame; IPR_Post = instantaneous pulse rate post-videogame; PANASPost_Pos = positive affect after game play; PANASPost_Neg = negative affect after game play; IPR_Diff = difference between pre-game and post-game instantaneous pulse rate; PANASPos_Diff = difference between pre-game and post-game positive affect; PANASNeg_Diff = difference between pre-game and post-game negative affect.

**Table 2 healthcare-12-00772-t002:** Correlations.

Variable		1	2	3	4	5	6	7
1. Age	*r*	—						
2. Sex	*r*	−0.01	—					
3. IGD_10	*r*	−0.21	−0.05	—				
4. PSS	*r*	0.02	0.32 *	0.03	—			
5. IPR_Diff	*r*	0.04	−0.07	−0.15	0.22	—		
6. PANASPos_Diff	*r*	−0.11	0.19	−0.09	0.10	0.04	—	
7. PANASNeg_Diff	*r*	−0.33 *	−0.07	−0.07	−0.25	0.33 *	0.98	—

* *p* < 0.05. Note: see Table 1 for abbreviations.

## Data Availability

Data supporting the reported results are available upon request.

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
