# Peer review of "Two Sides of the Same Virtual Coin: Investigating Psychosocial Effects of Video Game Play, including Stress Relief Motivations as a Gateway to Problematic Video Game Usage"

_healthcare, 2024, doi:10.3390/healthcare12070772_

Round 1

Reviewer 1 Report

Comments and Suggestions for Authors

I recently had the opportunity to review this manuscript and found the introduction particularly engaging and well-written. However, I have some concerns regarding the alignment between the research questions posed in the literature review and the empirical study conducted. When the authors wrote about the possible positive aspects of video game consumption, they mentioned increased connectivity, improved psychological well-being, and cooperation between players. When the authors wrote about the negative aspects, they mentioned, for instance, the escapism-driven use of video games, the addictive nature of playing these games. These are all factors which cannot be tested in the current research design. Ultimately, video games can be used to potentially regulate stress, emotions, moods, etc., and it is problematic for individuals who have problems with emotional regulation. They can possibly adopt maladaptive coping strategies. The authors also wrote about compensatory mechanisms, using a theory by Kardefelt-Winther. Based on a literature review such as this, I expected empirical research closely tied to the multifaceted nature of video game consumption. Surprisingly, then, the empirical study is a small test of whether playing a fun game for 20-30 minutes reduces or increases stress, measured both at the biological and self-reported levels, and positive and negative affect. Less surprisingly, playing the fun game led to decreased levels of stress and negative affect. What I fail to see is how this connects to the questions raised in the literature review. Having a drink after work might reduce stress levels and negative affect, however, it is very different from having an alcohol use disorder. Playing a card game for fun might also, but being addicted to gambling is not. What I mean is, I fail to see how the results of this small experiment can be generalized and reflect on the original problems mentioned in the literature review.

The authors used a self-report measure of problematic gaming, but the design of the study is extremely limited. Even individuals with high scores on IGD were invited to participate and did not play the game as their decision, so they did not play this particular game for any of the reasons mentioned in the literature review (e.g., as a compensatory mechanism).

The authors wrote: "It is expected that those scoring highly on problematic gaming behaviors will exhibit a higher affect shift." But why? This is not a game they chose to play, and they might turn to different games, different playing lengths, etc., to have the desired affect shift.

Additionally, the statement, "Regardless of motivations for play, we expect to find that video game play has a positive physiological impact on players," is perplexing to me, especially considering the complex picture the authors have drawn in the introduction.

It might have just skipped my attention, but I don't see a justification for H5.

2.2.5. Video Game Conditions... Isn't there only one condition, which is playing the game with a before and after measurement of the study's variables?

Also, I found it problematic to include participants with differing levels of experience. Some of the differences might be from the novel experiences those participants had who had no experience in how to play. This novelty can also lead to positive outcomes (or anxiety, or whatever). So, this is not only a comparison of gaming individuals with high IGD scores vs. low IGD scores but individuals who have no experience whatsoever.

The authors stated, "Self-reported levels of gaming disorder were the independent variable, whereas positive affect, negative affect, and a biological measure of stress (PPG) were the dependent variables." However, it should be noted that low levels of gaming disorder could also include non-players. For example, low scorers might not have experienced serious problems at work or school due to gaming either because they do not play at all, or because they manage to play in adaptive ways.

I would include some descriptive information (correlation table, means, standard deviations). I'm particularly interested in the mean levels and standard deviations of IGD scores.

Also, I completely agree with the authors that H4 and H5 cannot be investigated with such a small sample despite the a priori power analysis ("One possible explanation for this is that the data set used for analysis did not have sufficient power to identify small effect sizes."). Importantly, the a priori analysis investigated power concerning the differences between two dependent means (matched pairs), but the inclusion of IGD scores complicate the analysis.

I agree with all the limitations the authors mentioned in their paper (easiness of the game, lack of control group, etc.). Overall, I think that the paper is well-written, but its scientific contribution is questionable. 

Reviewer 2 Report

Comments and Suggestions for Authors

The paper faces an interesting topic employing a study that used a repeated-measures experimental design to investigate the relationships between stress, video gaming, and problematic video gaming behaviours.

It is based on an appropriate methodology even if it includes a convenience sample of 40 students.

As to the procedure, the study is based on Mario Kart, but the authors should better explain which is the plot, and which kind of behaviours it can elicit, it is a kart racing game, thus it elicits a sense of flow. If yes it can depend on the expertise of the gamers. Has it been measured? And the perceived level of expertise, and sense of difficulty (Authors pointed out that 'Participants reported a moderate level of gaming experience varying from no experience to experienced players of video', but if they put this variable how and if results change? Are there any gender differences?

In this regard, the authors can consult, for comparison, other studies that highlight the effects on the promotion of health and safety behaviors through video games such as D'Errico, F., et al. (2022). Scare-away risks: the effects of a serious game on adolescents' awareness of health and security risks in an Italian sample. Multimodal Technologies and Interaction, 6(10), 93., from which particular gender differences emerge. Can you deepen this aspect?

Furthermore, it could be interesting to highlight how gestures, voice and other body parameters also contribute to expressing emotions to be controlled during video games.

(see in general the works of Poggi, I … (2010). Cognitive modeling of human social signals. In Proceedings of the 2nd international workshop on Social signal processing (pp. 21-26).) as a theoretical basis for identifying bodily signals as social signals.

In general, the study is well conducted, but the role of the variables associated with the sample should be better explained and the limitations associated with the evaluation measures should be highlighted.

Reviewer 3 Report

Comments and Suggestions for Authors

This is an interesting work on a contemporary topic whose study has important implications for the design of prevention and intervention programs. Among the strengths of the work, I would highlight its careful writing (everything is comprehensible), its clarity and adequate structure throughout its development. Specifically, the introduction gathers very well the nature and scope of Video Gaming, in addition to approaching its boundaries, epicenters and associated symptomatology; the status quo of the literature that constitutes the foundation of this research proposal appears well documented. The hypotheses are also clear. In the Methods section there are, in my opinion, some weaknesses that significantly undermine the robustness of the findings and, consequently, the conclusions of the study. The absence of a control group (also pointed out by the authors as a limitation) does not allow us to conclude unequivocally that the changes are due to the influence of the video game and not to other factors. The playing time, and despite what some (although few) say about low times, perhaps should be longer to increase the probability of inducing statistically significant changes in stress-relief or mood regulation. In the absence of a control group and with such low times, serious doubts really arise.  Be that as it may, and in spite of these undeniable weaknesses, the results are well developed, the discussion responds to what is expected (the authors suggest possible explanations to the findings); I do not really understand the usefulness of generating section 4.1 (Interpretations of Results) since its content is still a discussion of findings.  The conclusions could be improved a little, but what does seem necessary is a greater development of the implications of the work.

Reviewer 4 Report

Comments and Suggestions for Authors

please see Attachment

Round 2

Reviewer 2 Report

Comments and Suggestions for Authors

Just some last typos on added references 

please check them (for instance Poggi and Francesca instead of Poggi and D'Errico F. ) 

The integrations have improved the study's limitations. 

Comments on the Quality of English Language

ok 

Author Response

We thank the reviewer for their comment - this has been corrected (see lines 553/554) - and for their positive feedback regarding the amendments to the transcript.